



# Dislocation creep and glide in experimentally deformed glaucophane aggregates

Lonnie J. Hufford[1], Leif Tokle[1], Whitney M. Behr[1], Luiz F. G. Morales[1,2], and Claudio Madonna[1]

[1]Structural Geology and Tectonics Group, Geological Institute, Department of Earth Sciences, ETH Zürich, Zürich, Switzerland

[2]Scientific Center for Optical and Electron Microscopy (ScopeM), ETH Zürich, Otto-Stern-Weg 3, 8093, Zürich, Switzerland

**Corresponding authors:** Leif Tokle (leif.tokle@eaps.ethz.ch) and Whitney M. Behr (wbehr@ethz.ch)

**Abstract.** Glaucophane is a key rheology-controlling mineral in the oceanic crust of subducting slabs at blueschist facies conditions. Studies of naturally deformed glaucophane suggest dislocation-related deformation mechanisms can be activated at some pressure-temperature-stress conditions in subduction environments; however, the strength of glaucophane deforming via these mechanisms remains unconstrained. To address this, we conducted load stepping experiments using a Griggs apparatus at temperatures of 600-700°C, 1.0 GPa, and shear strain rates of $\sim 1.2 \times 10^{-8} \text{s}^{-1}$ to $\sim 1.2 \times 10^{-3} \text{s}^{-1}$, with a starting grain size of $<63 \mu\text{m}$. The mechanical data from these experiments show a transition in the stress exponent from $2.8 \pm 0.2$ at relatively low stresses, indicative of dislocation creep, to 14-19 at relatively high stresses, consistent with dislocation glide. Microstructural analyses show kinking, undulose extinction, and a shift from sharp linear grain boundaries in the hydrostatic samples to more rounded and lobate sutured grain boundaries in the deformed samples. High internal misorientations (subgrains, undulose extinction) in both relict and fine-grained regions of the deformed samples further support the activation of dislocation-related mechanisms. Based on these observations, we develop flow laws for dislocation creep $n = 3$, $Q = 450 \pm 15$ kJ/mol, $A = 2.32 \times 10^{10}$ $MPa^{-n} \ s^{-1}$) and dislocation glide ($Q = 899 \pm 43$ kJ/mol, $C = 1.83 \times 10^{32}$, and $\alpha = 0.0123$). Extrapolations of our flow laws to geologic conditions suggests that dislocation glide is unlikely to occur at steady state conditions, while dislocation creep dominates at temperatures above 450°C at relatively large grain sizes of $\sim$1 mm or larger. These insights refine our understanding of glaucophane rheology and its implications for subduction zone mechanics.

## 1 Introduction

Mafic oceanic crustal rocks that occupy the top several kilometers of subducting oceanic lithosphere exert an important influence on the rheology, stress distributions, and earthquake potential of the subduction interface (e.g. Wang and Bilek, 2011; Agard et al., 2016; Sun et al., 2020; Phillips et al., 2020; Tulley et al., 2020; Ikari et al., 2020; Braden and Behr, 2021). As mafic oceanic crust is subducted, it enters the blueschist facies and undergoes important mineralogical changes, marked by abundant growth of sodic amphibole, along with possible lawsonite, epidote, white mica, quartz and other accessory phases (Evans, 1990). Despite that blueschist facies mafic rocks occupy a large portion of the downgoing slab and the subduction interface



itself, the rheological properties of blueschists remain poorly characterized. Observations from the rock record provide some
25 clues on the rheology of mafic blueschists relative to other potential subduction inputs. For example, blueschists are commonly
found as lenses or boudins within metasedimentary or serpentinite-matrix melanges (Liou et al., 1975; Cowan, 1978; Hefferan
et al., 2002; Ernst, 2016; Agard et al., 2018), while in mafic-dominated terranes metamorphosed to eclogite facies, blueschists
tend to form the matrix surrounding stronger eclogite blocks (Davis and Whitney, 2006; Cao et al., 2013; Kotowski and Behr,
2019). These observations suggest a qualitative rheological hierarchy in which the strength of mafic blueschists is intermedi-
30 ate between metasedimentary rocks and eclogite, but more quantitative constraints on the mechanisms controlling blueschist
deformation are needed.

Microstructural observations from naturally deformed sodic amphibole, as the primary rheologically-controlling mineral in
blueschists, suggest that a wide range of deformation mechanisms are possible, ranging from brittle/frictional creep, to grain
size sensitive and insensitive crystal-plastic (viscous) deformation mechanisms. Several studies, for example, suggest that
sodic amphibole deforms via dislocation mechanisms such as dislocation glide or creep, based on the presence of undulose
extinction, subgrain development, and/or dynamic recrystallization (Reynard et al., 1989; Kim et al., 2013; Cao et al., 2013;
Kim et al., 2015; Behr et al., 2018; Kotowski and Behr, 2019). Other studies discuss evidence for dissolution-precipitation
creep processes based on the presence of microboudinage with new amphibole growth in boudin necks, or new amphibole
overgrowths oriented parallel to the foliation (Misch, 1969; De Caroli et al., 2024). Some studies have also documented brittle
deformation and cataclasis (Ildefonse et al., 1990; Muñoz-Montecinos et al., 2023). Despite these insights, an experimental
framework to systematically quantify the influence of key variables such as temperature, pressure, strain rate, and grain size,
on these deformation processes has not yet been established.

Previous experimental work on sodic amphibole has focused on its bulk modulus (Jenkins et al., 2010), development of
crystallographic preferred orientations (CPO) (Park et al., 2020; Park and Jung, 2022), its seismic velocity and anisotropy
(Ha et al., 2019; Park and Jung, 2022), and the role of dehydration embrittlement (Kim et al., 2015; Incel et al., 2017). Tokle
et al. (2023b) provided the first flow law for mafic polymineralic blueschists, describing conditions under which they deform
via diffusion creep associated with microboudinage. In this paper, we build on these existing experimental constraints by
focusing on the mechanical and microstructural properties of sodic amphibole (glaucophane) aggregates. We present a suite
of load stepping deformation experiments in the general shear geometry conducted using a Griggs apparatus at temperatures
from 600 to 700°C and a pressure of $1.0 \pm 0.1$ GPa. We document a transition from dislocation creep at lower stresses to
dislocation glide at higher stresses, supported by stress exponent values ranging from $2.8 \pm 0.2$ to $19 \pm 1.0$, and microstructures
showing typical dislocation creep textures such as undulose extinction, subgrain development, sutured/lobate grain boundaries,
and recrystallized grains. We discuss the implications of these findings for the strength and rheology of mafic blueschists,
emphasizing the role of dislocation-related deformation mechanisms in strain accommodation and rheological heterogeneity
at depth in subducting slabs.





## 2 Methods

### 2.1 Starting Material and Experimental Approach

The starting material for our experiments is derived from a MORB blueschist from the Cycladic Blueschist Unit on Syros Island, Greece (same locality as described in Behr et al. (2018)). We obtained powders with >98% glaucophane, where large

glaucophane grains were sieved and crushed in a mortar and pestle to produce a fine-grained powder (<63 $\mu$m). All experiments were performed with the Griggs apparatus in the Rock Physics and Mechanics laboratory at ETH Zürich. The experiments used a solid salt (NaCl) assembly and were conducted "as-is" (without pre-drying or adding water to the sample). The samples were deformed in a general shear geometry with grooved alumina (Al$_2$O$_3$) pistons cut at 45°.

The experiments included one hydrostatic and five load stepping experiments all at a confining pressure of $\sim 1.0 \pm 0.1$

65 GPa. The hydrostatic experiment was brought to and kept at 700°C and 1.0 GPa for 24 hours. This was used to determine the starting grain size after pressurization and compaction as well as the initial microstructure prior to deformation. Load stepping experiments were carried out at 600°C, 650°C, 675°C, and 700°C with up to 14 load steps per experiment and shear strain rates ranging from $\sim 1.2 \times 10^{-8} \mathrm{s}^{-1}$ to $\sim 1.2 \times 10^{-3} \mathrm{s}^{-1}$. Figure 1a shows the mechanical data for the 700°C load stepping experiment. Each step achieved a mechanical steady state where at least 20 $\mu$m of axial displacement was accumulated before

70 changing to the next load step. The shear strain rate for each load step was calculated from the linear portion of the shear strain data with respect to time, while the shear stress was taken from the steady state portion of each step (Fig. 1a). Several load steps did not accumulate enough strain and are excluded from the final data set. Shear stresses and shear strain rates are converted to equivalent stresses

$$\sigma_{equiv} = 2\tau$$

and equivalent strain rates

$$\dot{\varepsilon}_{equiv} = \frac{\dot{\gamma}}{\sqrt{3}}$$

after Paterson and Olgaard (2000). Each experiment achieved a total shear strain of $\sim$0.8. After the experiments, samples were cut in half perpendicular to the shear plane where one half was used for thin sectioning. The 650°C experiment localized during the last load step and the sample accumulated too much strain to evaluate the microstructures. One of the motivations

for conducting load stepping experiments was to suppress brittle/cataclastic behavior observed at peak stress conditions in constant rate deformation experiments (see companion paper, Hufford et al. (submitted concurrently)).

### 2.2 Data Processing

Force data were collected by an external load cell on the Griggs apparatus and recorded at a sampling rate of 1 Hz. A modified version of the open source code RIG (https://mpec.scripts.mit.edu/peclab/software/) for the ETH Zürich Griggs apparatus was

85 used to process the mechanical data. Raw data are down-sampled and a median filter is applied to reduce electrical noise. The





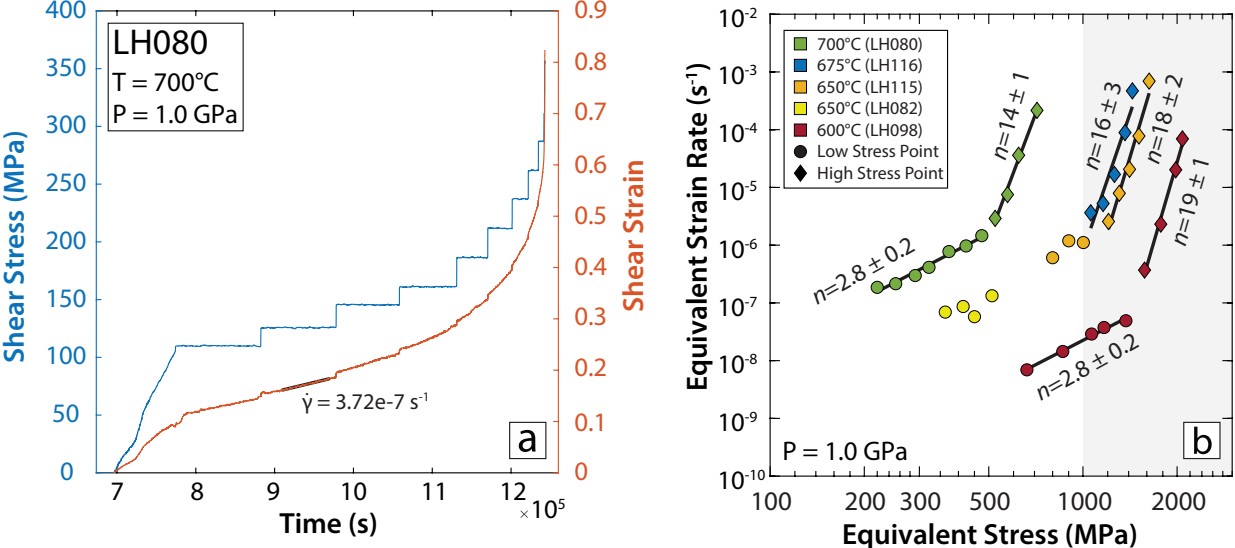

**Figure 1.** Mechanical data of the load stepping experiments. a) The mechanical data for load stepping experiment LH080 conducted at 700°C and 1.0 GPa showing 11 deformation steps. The slope of the shear strain versus time plot represents the shear strain rate as illustrated by the black line in the second deformation step. b) Plot of equivalent stress versus equivalent strain rate showing the steady state mechanical data from the different load stepping experiments. The grey shaded region represents stresses above the Goetze criterion. Uncertainties in $n$ represents one standard deviation.

mechanical data were corrected for sample thinning and area correction. Mechanical data for all experiments can be found in Table 1.

## 2.3 Microstructural Analyses

Backscattered electron (BSE) images and crystallographic orientation data were collected using a scanning electron microscope
(SEM) equipped with an electron backscatter diffraction (EBSD) detector at the Scientific Centre for Optical and Electron Microscopy (ScopeM) of ETH Zürich. All samples were carbon coated before imaging. BSE and EBSD data were collected with a Hitachi SU5000 FEG SEM, an Oxford Symmetry 2 EBSD camera and Aztec software. Details on sample polishing (Table S1) and run settings are provided in the supplementary material. All EBSD maps from deformed samples were made in the central regions of the sample. Mis2mean maps show intragranular misorientations, calculated as the minimum rotation
angle between the pixel orientation and the mean orientation of the grain (Chauve et al., 2017). Optical photomicrographs were also taken to document the microstructures in cross-polarized light.



**Table 1.** Mechanical Data

| Sample ID | Shear Stress (MPa) | Equivalent Stress (MPa) | Shear Strain Rate (s$^{-1}$) | Equivalent Strain Rate (s$^{-1}$) | T (°C) | P (GPa) | Starting Grain Size ($\mu$m) |
|---|---|---|---|---|---|---|---|
| LH080 | 110 | 220 | 3.22E-07 | 1.86E-07 | 700 | 1.05 | Fine: <63 |
| | 126 | 252 | 3.72E-07 | 2.15E-07 | 700 | 1.05 | |
| | 146 | 291 | 5.16E-07 | 2.98E-07 | 700 | 1.05 | |
| | 161 | 322 | 7.15E-07 | 4.13E-07 | 700 | 1.05 | |
| | 187 | 373 | 1.34E-06 | 7.75E-07 | 700 | 1.05 | |
| | 212 | 423 | 1.68E-06 | 9.69E-07 | 700 | 1.05 | |
| | 238 | 475 | 2.53E-06 | 1.46E-06 | 700 | 1.05 | |
| | 262 | 524 | 5.04E-06 | 2.91E-06 | 700 | 1.05 | |
| | 287 | 574 | 1.30E-05 | 7.51E-06 | 700 | 1.05 | |
| | 312 | 623 | 6.25E-05 | 3.61E-05 | 700 | 1.04 | |
| | 357 | 713 | 3.74E-04 | 2.16E-04 | 700 | 1.04 | |
| LH082 | 182 | 363 | 1.20E-07 | 6.93E-08 | 650 | 1.05 | Fine: <63 |
| | 207 | 414 | 1.50E-07 | 8.66E-08 | 650 | 1.05 | |
| | 225 | 450 | 9.99E-08 | 5.77E-08 | 650 | 1.05 | |
| | 256 | 512 | 2.30E-07 | 1.33E-07 | 650 | 1.05 | |
| LH089* | - | - | - | - | 700 | 1.01 | Fine: <63 |
| LH098 | 331 | 662 | 1.20E-08 | 6.93E-09 | 600 | 1.03 | Fine: <63 |
| | 431 | 862 | 2.49E-08 | 1.44E-08 | 600 | 1.03 | |
| | 533 | 1066 | 5.01E-08 | 2.89E-08 | 600 | 1.03 | |
| | 584 | 1168 | 6.50E-08 | 3.75E-08 | 600 | 1.02 | |
| | 686 | 1372 | 8.50E-08 | 4.91E-08 | 600 | 1.02 | |
| | 787 | 1574 | 6.41E-07 | 3.70E-07 | 600 | 1.02 | |
| | 887 | 1774 | 4.00E-06 | 2.31E-06 | 600 | 1.02 | |
| | 990 | 1980 | 3.50E-05 | 2.02E-05 | 600 | 1.02 | |
| | 1040 | 2080 | 1.20E-04 | 6.93E-05 | 600 | 1.01 | |
| LH115 | 400 | 800 | 1.05E-06 | 6.06E-07 | 650 | 1.02 | Fine: <63 |
| | 451 | 901 | 2.06E-06 | 1.19E-06 | 650 | 1.02 | |
| | 502 | 1003 | 1.92E-06 | 1.11E-06 | 650 | 1.02 | |
| | 604 | 1207 | 4.45E-06 | 2.57E-06 | 650 | 1.01 | |
| | 654 | 1307 | 1.37E-05 | 7.92E-06 | 650 | 1.01 | |
| | 705 | 1409 | 3.57E-05 | 2.06E-05 | 650 | 1.01 | |
| | 755 | 1510 | 1.35E-04 | 7.77E-05 | 650 | 1.01 | |
| | 814 | 1628 | 1.20E-03 | 6.93E-04 | 650 | 1.01 | |
| LH116 | 530 | 1059 | 6.30E-06 | 3.64E-06 | 675 | 1.03 | Fine: <63 |
| | 580 | 1160 | 9.13E-06 | 5.27E-06 | 675 | 1.03 | |
| | 630.5 | 1261 | 2.91E-05 | 1.68E-05 | 675 | 1.03 | |
| | 682 | 1363 | 1.54E-04 | 8.90E-05 | 675 | 1.03 | |
| | 720 | 1439 | 8.14E-04 | 4.70E-04 | 675 | 1.03 | |

*Hydrostatic sample (LH089): time at $T$ and $P$ for hydrostatic conditions= 24 hrs.





# 3 Results

## 3.1 Hydrostatic Microstructures

The hydrostatic experiment shows crushing at grain margins, kinking, and fracturing related to compaction (Fig. 2a). Figure
2a shows a weak sample-scale foliation and a weak SPO, where the long axis of the grains aligns sub-parallel to the shear
plane, likely due to the alignment of tabular grains during compaction. This has been observed in other experimental studies on
tabular minerals (Tokle et al. (2023b) and Hufford et al. (submitted concurrently)). Most grains are angular in shape regardless
of grain size (Figure 2d). Based on the inverse pole figure map oriented in the x-direction (IPFX), many grains have the [001]
and [010] axes oriented parallel to the shear plane (Figure 3a). Other grains display intragranular misorientations observed
with the EBSD Mis2Mean map (Figure 3b), that are likely inherited from the starting material and/or produced by compaction
during pressurization.

## 3.2 Mechanical Data

The load stepping experiments show two stress exponent slopes associated with smaller and larger values, which coincide with
increasing stress. A distinct change in slope is observed for the 600 and 700°C mechanical data where at lower stresses, the
stress exponents are 2.8± 0.2, while at higher stresses, the stress exponents increase to 14-19 (Fig. 1b). Linear fits to the data
were chosen based on the point of greatest change in the stress exponent magnitude (Fig. 1b). A fit was not applied to the 650°C
mechanical data as it does not achieve a similar range in stress consistent with the 600 and 700°C mechanical data, however it
does show a similar trend where the mechanical data show an increase in the stress exponent magnitude with increasing stress
(Fig. 1b). The stresses achieved in the 600, 650, and 675°C load stepping experiments exceed the Goetze criterion (Fig. 1b);
however, this does not appear to influence the mechanical properties of the samples, where the low and high stress exponent
relationships are observed above and below the Goetze criterion for different temperatures (Fig. 1b). This is consistent with
previous deformation experiments on amphibole-bearing samples (Okazaki and Hirth, 2020).

## 3.3 Deformation Microstructures

Due to the small amount of strain in the load stepping experiments ($\gamma\sim$0.8), sample-scale foliations are not significantly
different from the starting microstructure with no strain localization found within the samples (Fig. 2). A weak SPO, based on
the alignment of the long axis of the grains is observed in all samples (Fig. 2). The average grain size is consistent throughout
all experiments, except for the 700°C deformed sample, which had a larger average grain size than the other four deformed
samples. Despite these similarities with the hydrostatic experiment, the grain boundaries in the deformed samples are less
angular/more rounded and lobate, especially in the finer-grained regions (Figs. 2e and f and 3d). Undulose extinction is optically
identified in all samples based on broad sweeping extinction contrasts in relict grains (Fig. 2e and f, white arrows), while kink
bands are also identified by the presence of sharp linear extinction contrasts across grain interiors (Fig. 4, red arrows). Subgrains
are also identified in the 700°C sample (LH080), both optically and in the mis2mean EBSD map, where the misorientation is



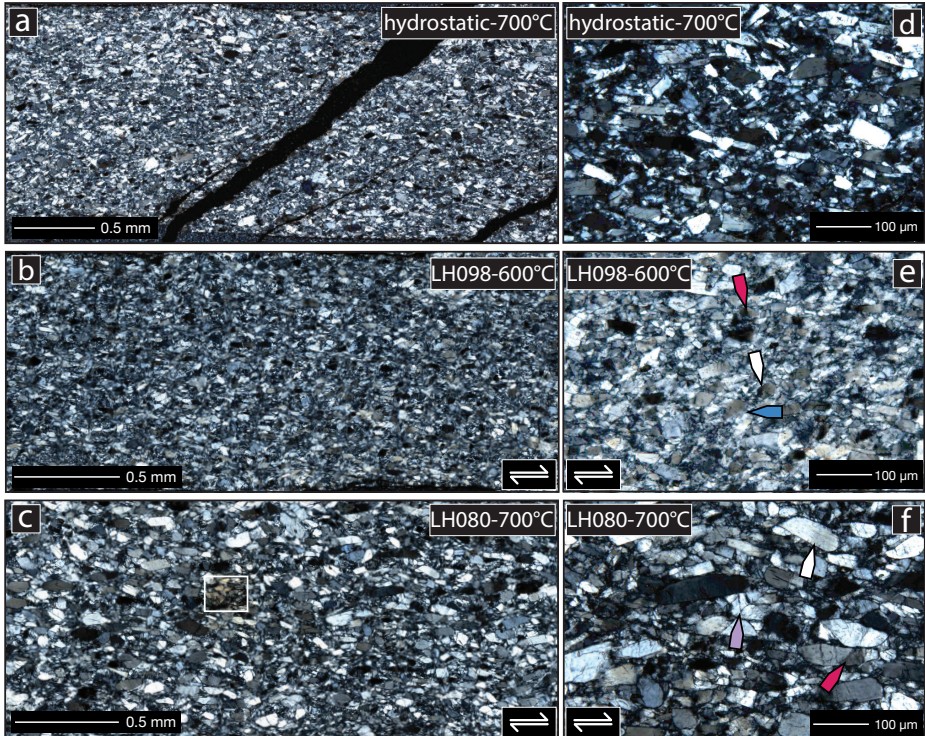

**Figure 2.** Cross-polarized light photomicrographs of the hydrostatic and load stepping samples. Sample-scale photomicrographs of a) hydrostatic sample LH089, b) load stepping sample LH098, and c) load stepping sample LH080. The black spaces in a) at 45° are the result of decompression cracking. The white box in c) indicates the location of the EBSD analysis conducted for Fig. 3. Zoomed-in photomicrographs of d) hydrostatic sample LH089, e) load stepping sample LH089, and f) load stepping sample LH080. The blue arrow in e) highlights subgrains. The purple arrow in f) highlights fractures. For e,f the white arrows indicate undulose extinction and red arrows highlight kinked grains.

unrelated to microfractures (Figs. 3d and 2f, blue arrow). Minor amounts of grain fracturing, commonly along cleavage plains, are observed with minimal shear offset (Fig. 2 f, purple arrow). The IPFX map from the 700°C sample shows a spread of

orientations of the relict grains with many having their [001] or [010] axes oriented in the direction of shear (Fig. 3c). Based on the IPFX and mis2mean maps, these oriented relict grains typically show higher amounts of internal misorientations and subgrain development relative to relict grains with other orientations (Fig. 3d and e).

Hufford et al. (*submitted companion manuscript*), shows breakdown reactions where the original sodic amphibole reacts to produce albite and a compositionally new amphibole. Based on the BSE photomicrographs from the 600 and 700°C load-

stepping experiments, there are minor amounts of glaucophane breakdown (<1% of the imaged area) (Fig. 4); however, the small amount of albite and compositionally new amphibole is not interconnected and there is no strain localization within the reacted regions relative to the surrounding sodic amphibole (Fig. 4). Therefore, we interpret albite and the compositionally new amphibole to have no effect on the mechanical or microstructural evolution of the deformed samples.



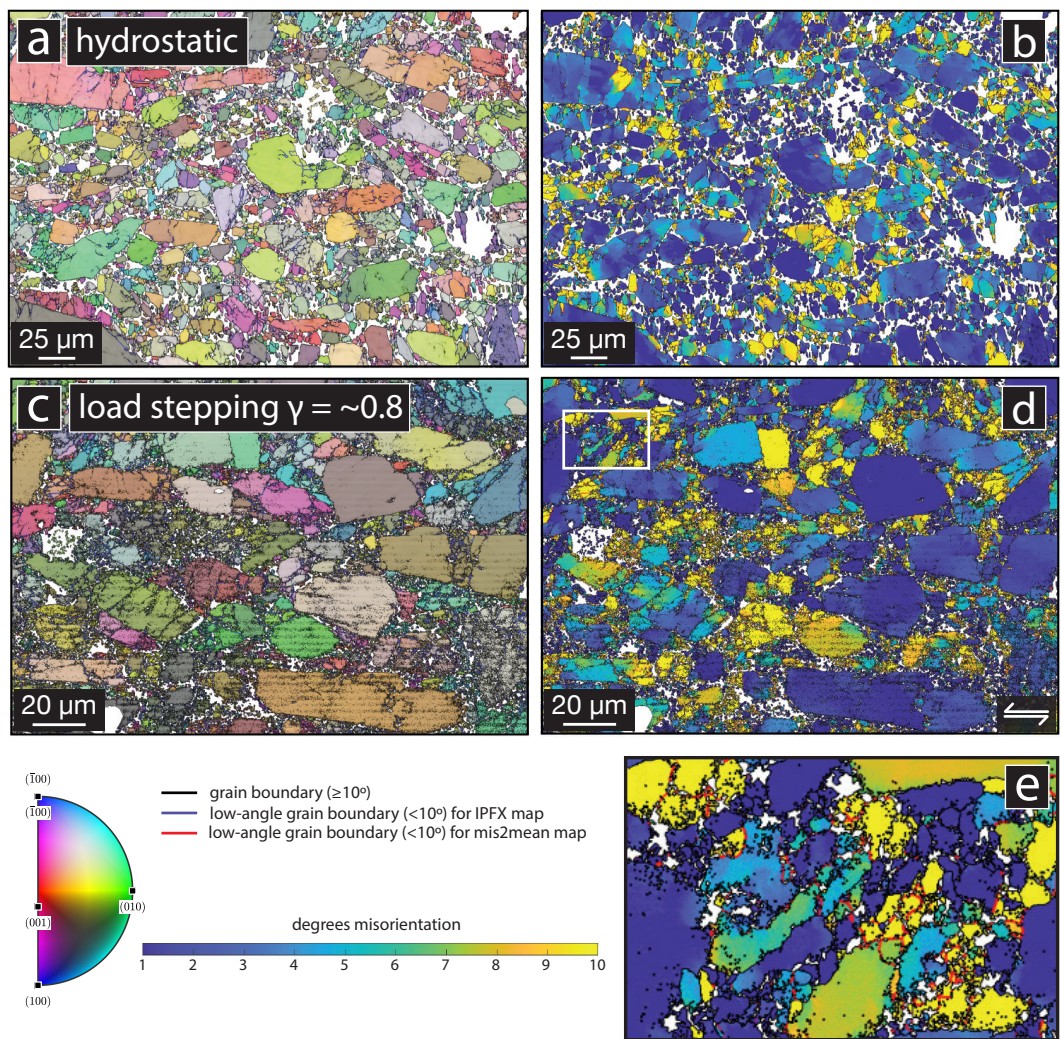

**Figure 3.** EBSD maps of the a,c) the hydrostatic sample LH089 and b,d) the load stepping sample LH080. The EBSD maps on the left (a,c) are inverse pole figure maps in the x-direction (IPF-X) and the maps on the right (b,d) are mis2mean maps. The white box in d) represents the location of e). The shear plane is parallel to the horizontal plane of each EBSD map.




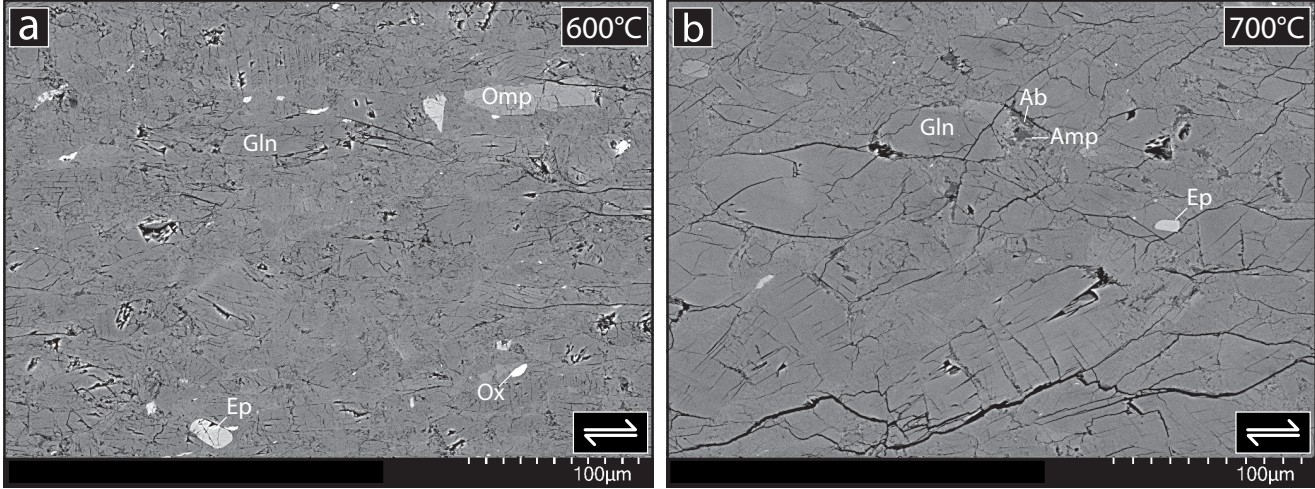

**Figure 4.** BSE micrographs of the 600 (LH098) and 700°C (LH080) load stepping samples. a) No evidence for glaucophane breakdown reaction. b) Isolated pocket amounts of grain size reduction and glaucophane breakdown where albite (Ab) and a compositionally new amphibole (Amp) are observed. Ox - oxide, Ep - Epidote, Omp - omphacite, and Gln - glaucophane. The shear plane is parallel to the horizontal plane of each BSE micrograph.

## 4 Discussion

### 4.1 Interpreted Deformation Mechanisms

Based on the deformation microstructures and corresponding mechanical data, we interpret our samples to have deformed by dislocation creep at low stresses and dislocation glide at higher stresses. The load stepping samples described in Section 3 show microstructural evidence for undulose extinction, subgrain development, and lobate grain boundaries, which are consistent with dislocation-related deformation mechanisms (Hirth and Tullis, 1992). Additionally, the change in grain shape between the angular hydrostatic microstructures and the more rounded and lobate deformed microstructures implies the movement of grain boundaries, which can also be attributed to dislocation activity via grain boundary migration recrystallization (Hirth and Tullis, 1992; Stipp et al., 2002; Platt and Behr, 2011). This interpretation is also supported by the mechanical data, where stress exponents for the 600 and 700°C load stepping experiments are $2.8 \pm 0.2$ (Fig. 1b) at low stress conditions, which is consistent with theoretical values for dislocation creep (Poirier, 1985). The larger stress exponents at higher stresses (14-19) are consistent with power-law breakdown and a rheological transition from dislocation creep to dislocation glide (Frost and Ashby, 1982). Microstructurally, the large degree of intragranular misorientations, fracturing, and kinking in both the relict and finer grain sizes support this rheological transition, whereas, dislocation creep would typically produce finer grains with low internal strain (Fig. 3d and e).





## 4.2  Comparison to Previous Experimental Studies

Previous experimental work on amphibole and amphibole-bearing samples show a wide-range of deformation mechanisms and stress exponent relationships. Figure 5 compares temperature and stress exponent data for deformation experiments that were conducted at different confining pressures, equivalent stresses, and equivalent strain rates. The change in deformation mechanism correlates well with the deformation microstructures observed in each study. For stress exponents greater than 20, deformed amphibole-bearing greenschist and epidote amphibole schists show microstructures consistent with semi-brittle
flow, displaying a significant amount of intragranular fracturing and kinking (Okazaki and Hirth, 2020). The high stress segment of our load stepping experiments display stress exponents of 14-19, interpreted to deform by dislocation glide, where the microstructures show high internal stresses of the relict and recrystallized grains with minor amounts of kinking (Fig. 3d and e). Stress exponents of ∼3 are documented in (Hacker and Christie, 1990) and the low stress segment of our load stepping mechanical data (Fig. 5). Hacker and Christie (1990) deformed synthetic mixtures of amphibole and plagioclase, and observed
subgrain boundaries and localized recrystallization in amphibole along grain and phase boundaries, which were interpreted to represent evidence for dislocation creep. We observe similar microstructures in our load stepping samples such as subgrain development as well as changes in grain boundary morphology. Stress exponents of $\leq 2$ are documented in samples associated with diffusion creep (Fig. 5). Tokle et al. (2023b) deformed a powdered blueschist aggregate where the stress exponents ranged from 1.8-2.2 and microstructurally observed diffusion creep limited by microboudinage in sodic amphibole, where a composi-
tionally new sodic-calcic amphibole diffuses into the boudin neck. Deformation experiments on fine-grained amphibolite show a correlation between the alignment of amphibole SPO and CPO with stress exponents of $\sim$ 1.0-1.8, suggesting diffusion creep with oriented grain growth and rigid body rotation in amphibole (Getsinger and Hirth, 2014; Getsinger, 2015).

The specific grain-scale deformation processes that we observe in our experiments during steady state deformation have also been observed in other experimental studies on sodic amphiboles. General shear deformation experiments on a natural epidote
blueschist showed local misorientations indicative of dislocation activity in grains larger than 30 $\mu$m, which is consistent our observations (see Fig. 6 in Park et al. (2020)) (Figs. 3). Additional general shear deformation experiments on a lawsonite blueschist showed the development of subgrains and stacking faults in fine-grained glaucophane, suggesting dislocation glide as well as climb were active, also consistent with our observations (Kim et al., 2015).

## 4.3  Comparison to Observations from Natural Rocks

Analysis of naturally deformed sodic amphibole have previously documented evidence for dislocation activity. Reynard et al. (1989) examined slip systems in sodic amphibole from rocks in the western Alps deformed at relatively low (350-450°C) and high (550-600°C) temperatures. For the high temperature samples, Reynard et al. (1989) interpreted dislocation glide with active recovery processes (i.e., dislocation creep) based on transmission electron microscopy data, including the presence of well-organized subgrain boundaries, and evidence for dislocation activity on multiple slip systems. For the lower temperature
samples, microstructural evidence showed glide only on the (010)[001] slip system with no evidence for recovery, consistent with dislocation glide as the primary deformation mechanism. Similar evidence for dislocation activity has been documented





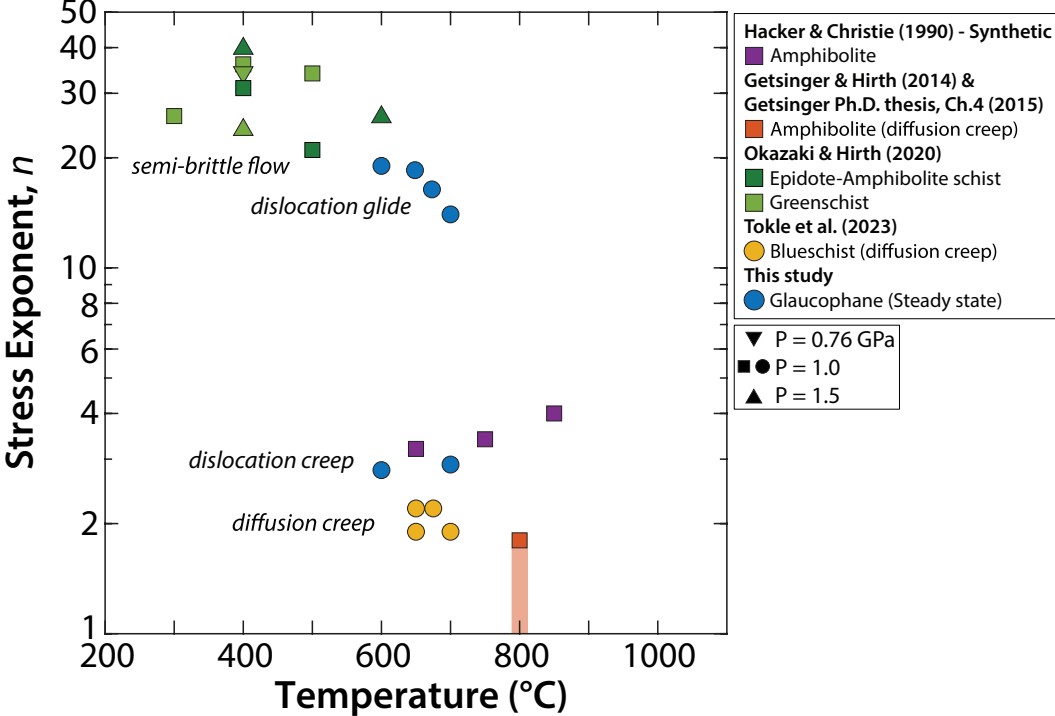

**Figure 5.** Plot of temperature versus stress exponent from deformation experiments on mafic schists, amphibolites, blueschists, and glaucophane. Comparison showing semi-brittle deformation associated with stress exponents >20, dislocation glide 14-19, dislocation creep ~ 3 and diffusion creep associated with lower stress exponents. (Hacker and Christie, 1990; Getsinger and Hirth, 2014; Okazaki and Hirth, 2020; Tokle et al., 2023b; Getsinger, 2015)

for glaucophane at or near eclogite-facies conditions on Syros Island (Greece) (Behr et al., 2018; Kotowski and Behr, 2019), the Diablo Range (USA) (Kim et al., 2013), and Corsica (France) (Choi et al., 2024).

### 4.4 Dislocation Creep and Glide Flow Laws

Based on our interpretation of both the mechanical and microstructural data, we use our load stepping experiments to constrain two flow laws describing the rheological behavior of glaucophane, one representing dislocation creep (occurring at lower stresses), and the other representing dislocation glide (occurring at higher stresses). The dislocation creep flow law parameters are determined using a least squares regression on the lower stress mechanical data (circles in Fig. 6), where the stress exponent for dislocation creep is assumed to be 3, which is within the uncertainty of our load stepping experiments and consistent with

theoretical values for dislocation creep (e.g., (Poirier, 1985)). The dislocation creep flow law equation is defined as,

$$\dot{\varepsilon} = A\sigma^n exp\left(\frac{-Q}{RT}\right)$$



where $\dot{\varepsilon}$ is strain rate in $s^{-1}$, $A$ is a material parameter in $MPa^{-n}s^{-1}$, $\sigma$ is the differential stress in MPa, $n$ is the stress exponent, $Q$ is the activation enthalpy in kJ/mol, $R$ is the gas constant, and $T$ is temperature in K. The least squares regression provides dislocation creep flow law parameters of $Q = 450 \pm 15$ kJ/mol and $A = 2.32 \times 10^{10}\ MPa^{-n}\ s^{-1}$.

The dislocation glide flow law parameters are determined using the modified exponential creep flow law described in Tokle et al. (2023a). The modified exponential creep flow law equation is defined as,

$$\dot{\varepsilon} = C\sigma^2 exp\left(\alpha\sigma - \frac{Q}{RT}\right)$$

, where $\dot{\varepsilon}$ is strain rate in $s^{-1}$, $C$ is a material parameter, $\sigma$ is the differential stress in MPa, $\alpha$ is the relationship between log strain rate versus stress, $Q$ is the activation enthalpy in kJ/mol, $R$ is the gas constant, and $T$ is temperature in K. Following,
Kronenberg et al. (1990), $\alpha$ is first calculated, where we determine a value of 0.0123 (Fig. S3). The least squares regression is fit only to the high stress data (diamonds in Fig. 6), where two glide flow laws are developed; one including all high stress data (600-700°C) (Fig. 6a) and the other excluding the 700°C data (600-675°C) (Fig. 6b). The glide flow law parameters for the fit to all high stress data are $Q = 1407 \pm 129$ kJ/mol and $C = 5.38 \times 10^{59}$, whereas the flow law parameters for the fit excluding the 700°C data are $Q = 899 \pm 43$ kJ/mol and $C = 1.83 \times 10^{32}$ (Fig. 6). All uncertainties are listed as one standard deviation. The
motivation to fit two different glide flow laws originates from the observation that the 700°C high stress mechanical data is notably weaker than the other high stress mechanical data at lower temperatures (Fig.s 1b and 6). In addition, the 700°C sample has a larger average grain size than the lower temperature samples, which all have a smaller, more comparable grain size (Fig. 2). This may suggest that the 700°C high stress mechanical data is weaker as a result of the larger grain size relative to the lower temperature experiments. This observation is consistent with recent deformation experiments on olivine aggregates, where at
low-temperature plasticity conditions (e.g., dislocation glide), the coarser-grained samples were weaker than the finer-grained samples (Hansen et al., 2019). If this relationship is accurately reflected in our mechanical data, it suggests that the transition between dislocation creep and glide is dependent on grain size. However, more experiments are required to further test and constrain this relationship. Given the more consistent grain sizes between the 600, 650, and 675°C samples, we prefer the glide flow law parameters that are fit without the 700°C data.

In comparison to previous experimental studies on amphibole-bearing samples, the activation enthalpy for both dislocation creep and glide is higher than the blueschist diffusion creep via microboudinage flow law ($Q = 384 \pm 15$ kJ/mol) by Tokle et al. (2023b) and the amphibolite diffusion creep flow law ($Q = 305$-$353$ kJ/mol) by Getsinger (2015). While the activation enthalpy for dislocation creep is consistent with the general range of geologic materials, the activation enthalpy for dislocation glide is notably higher. The elevated value may be related to several factors. First, if there is a grain size effect influencing the strength of the samples (e.g., Hansen et al. (2019)), this likely influences the activation enthalpy value we determine because the effect of grain size is ignored in our analysis. Second, is the potential effect of the sample stresses exceeding the Goetze criterion. At these conditions, deformation is more likely to involve brittle or semi-brittle mechanisms in addition to dislocation glide, complicating the interpretation of the measured activation enthalpy. However, as mentioned previously, exceeding the Goetze criterion does not appear to influence the bulk rheological properties and there is no obvious microstructural observations for




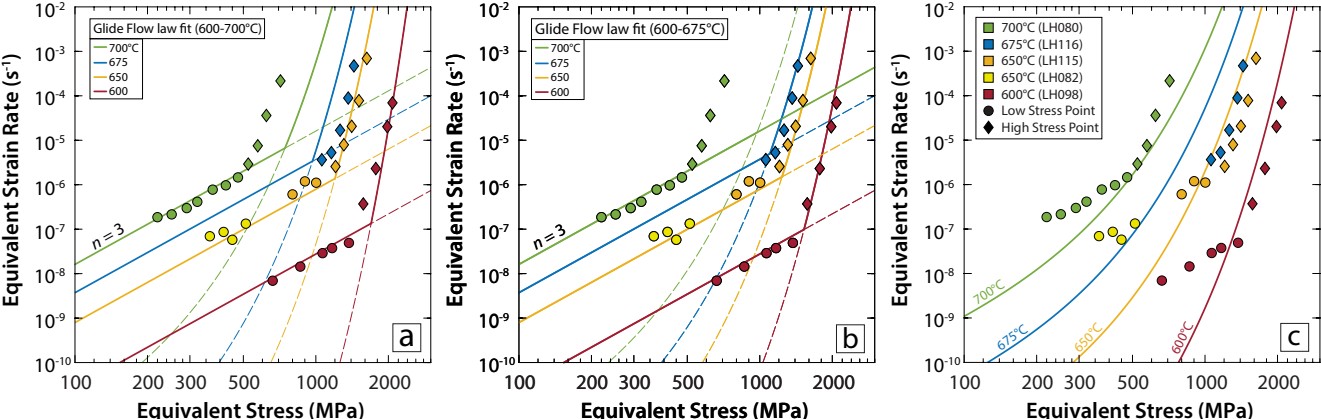

**Figure 6.** Plots comparing derived flow laws with the load stepping mechanical data. a) Dislocation creep flow law and a glide flow law derived from all high stress mechanical data (600-700°C). Solid colored lines represent the composite rheology, where the dashed lines represent the extrapolation of the end member flow laws. b) Dislocation creep flow law and a glide flow law derived from the 600, 650, and 675°C high stress mechanical data. The 700°C dislocation glide dashed line represents the predicted strength based on the derived flow law and not the fit to the 700°C high stress data. c) A single exponential creep flow law is fit to all mechanical data.

increased brittle deformation between the 700°C sample, which did not exceed the Goetze criterion and the 600°C sample that did exceed the Goetze criterion (Fig. 2). As mentioned previously, more experimental data is required to further constrain the rheological properties of dislocation glide in glaucophane.

Finally, we attempted to fit both the low and high stress mechanical data to a single exponential flow law; however, this did not provide a good fit to the mechanical data (Fig. 6c). This provides additional evidence supporting the interpretation of dislocation creep rather than glide at the low stress conditions (Fig. 6).

## 4.5 Deformation Mechanism Maps for Subduction Conditions

To investigate the applicability of our dislocation creep and dislocation glide flow laws, we plot deformation mechanism maps incorporating these flow laws alongside the sodic amphibole diffusion creep limited by microboudinage flow law. These maps illustrate the dominant deformation mechanisms across a range of stress, temperature and grain size conditions relevant to the subduction interface (Fig. 7).

At the lowest temperature of 350°C, diffusion creep is predicted to be the dominant deformation mechanism for sodic amphibole across the relevant range of stresses and grain sizes. However, the strain rates predicted for these low temperatures are much slower than would be expected for the subduction interface (e.g. Abila et al., 2024; Lamb, 2006; Alcock et al., 2005), which implies that activating creep mechanisms in sodic amphibole is unlikely at these conditions. Instead, at these conditions, mafic oceanic crust is more likely to deform via brittle mechanisms and/or subduction interface strain is more likely to be accommodated by surrounding rheologically weaker materials, such as metasediments (Fig. 7).



At higher temperatures of 450 and 550°C, diffusion creep is still the dominant deformation mechanism at most grain sizes and stresses, but dislocation creep begins to dominate at larger grain sizes (1-10 mm) and higher stresses ($\geq$ 10 MPa) (Fig. 7b and c). The predicted range of strain rates ($10^{-15}$ through $10^{-10}$/s) are more consistent with geologic estimates for subduction zone deformation. At 450°C, metasediments may still be rheologically weaker than glaucophane-bearing rocks, thus still potentially controlling deformation. However, at 550°C, the viscosities of glaucophane aggregates and metasediments may become more comparable, as observed in naturally deformed high pressure rocks from the subduction zone interface at the Fabrikas outcrop in Syros Island, Greece (Kotowski and Behr, 2019).

At the highest temperature of 650°C, diffusion creep continues to dominate at low stresses and small diffusion length scale/grain size, while dislocation creep takes over at larger grain sizes. Notably, dislocation glide is not predicted to occur at these conditions (Fig. 7). The limited field for dislocation glide to occur is available in Fig.s (S4 and S5). Our maps suggests that this mechanism is unlikely to occur under steady-state conditions at the subduction interface. Instead, dislocation glide may represent a transient deformation mechanism activated during high-stress events, such as localized stress concentrations or seismic loading.

Our deformation mechanism maps both support and challenge previously interpreted conditions for dislocation mechanisms in sodic amphibole. Reynard et al. (1989) analyzed naturally deformed blueschists from the Sesia-Lanzo Zone (Aosta Valley, Western Alps), where they document the activation of multiple slip systems, consistent with dislocation creep at 550-600°C. This interpretation aligns with our maps at these temperatures (Fig. 7c and d). However, the limited range of conditions where dislocation glide is accessible in Figures S4 and S5 shows that high temperatures, high stresses, and/or fast strain rates are necessary to access glide dislocation and conflicts with previously reported dislocation glide conditions in glaucophane (Reynard et al., 1989). Dislocation glide on the (010) [001] slip system has been shown to be easier for sodic amphibole than other amphibole varieties and has been interpreted to occur at temperatures as low as 350-450°C (Reynard et al., 1989). This discrepancy further supports our hypothesis that dislocation glide may only activate during high-stress, transient events rather than during steady-state deformation.

## 4.6 Implications for Subduction Interface Viscosity

Our dislocation creep flow law also has broader implications for the long term strength/viscosity of blueschist-facies oceanic crust relative to other potential subduction input materials and metamorphic conditions. Figure 8 shows viscosity versus temperature for a constant strain rate of $1 \times 10^{-12}$ s$^{-1}$ and a range of flow laws that represent e.g. sediments (quartz dislocation creep (Tokle et al., 2019)), blueschist-facies metamafic rocks (this study and the Tokle et al. (2023b) flow law), and eclogite-facies metamafic rocks (eclogite dislocation creep (Zhang and Green, 2007)). The two blueschist flow laws show intermediate viscosities between those for quartz and eclogite– this is consistent with the rheological hierarchy that would be inferred qualitatively from observations of block-and-matrix structures in subduction melanges (Davis and Whitney, 2006; Cao et al., 2013; Kotowski and Behr, 2019). For the strain rate shown in Fig. 8, the viscosity for blueschists does not approach that of quartz until ∼600°C, which is also the temperature expected for the complete breakdown of blueschist to form eclogite (Evans, 1990). Thus, blueschists are expected to remain strong relative to quartz-rich rocks over their entire stability field at this strain rate



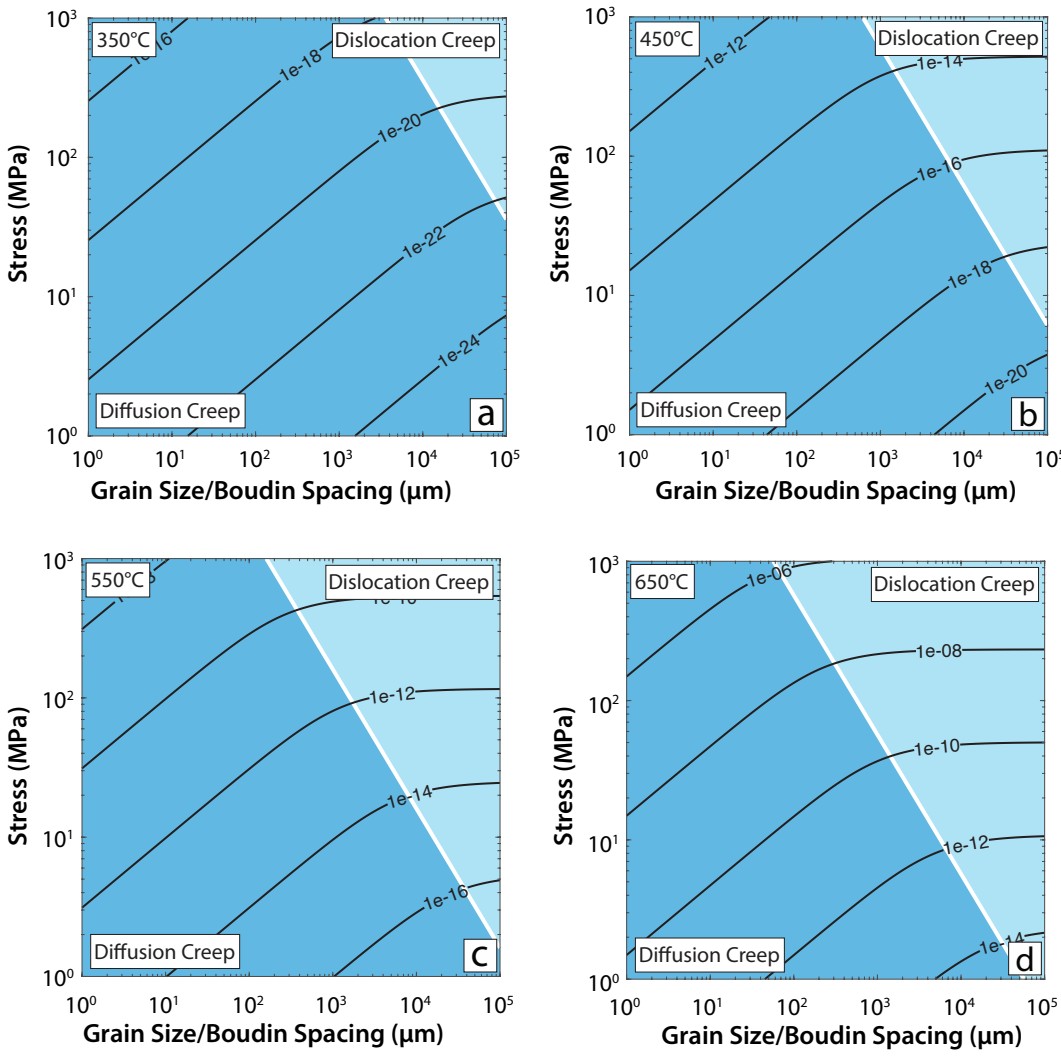

**Figure 7.** Deformation mechanism maps comparing stress and grain size/boudin spacing for 350, 450, 550, and $650°C$. The diffusion creep flow law is the sodic amphibole diffusion creep limited by microboudinage flow law from Tokle et al. (2023b). The code used to produce the deformation mechanism maps is modified from Warren and Hirth (2006).

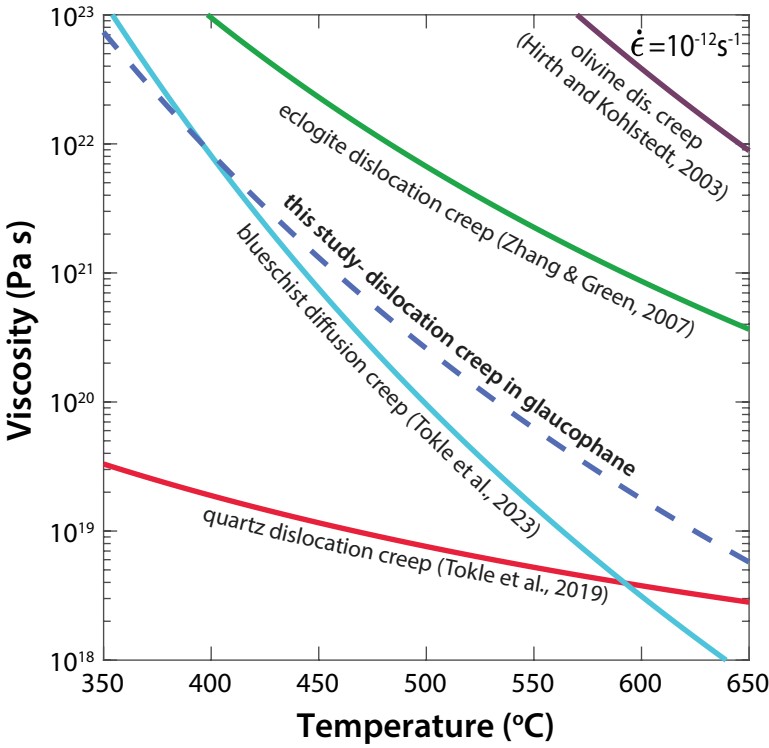

**Figure 8.** Plot of viscosity versus temperature at a constant strain rate of $1 \times 10^{-12}$/s comparing the strength of the glaucophane dislocation creep flow law with other strain-accommodating materials: quartz $n = 4$ dislocation creep (Tokle et al., 2019), blueschist diffusion creep with a boudin spacing of 100 $\mu$m (Tokle et al., 2023b), and eclogite (Zhang and Green, 2007). Olivine is included for comparison (Hirth and Kohlstedt, 2003).

and selected grain size/boudin spacing. The viscosity contrast between blueschist creep mechanisms and quartz dislocation creep is 1-2 orders of magnitude on average over the blueschist temperature range. A viscosity range of $\sim 10^{-18}$ to $10^{-20}$ Pa·s is consistent with the range expected to still permit steady-state subduction, but lower viscosities should promote faster subduction velocities and vice versa (Behr et al., 2022; Behr and Becker, 2018). Thus, the viscosities predicted by our flow

law support the suggestion that sediment- or serpentinite-rich subduction interfaces should be weaker than those dominated by mafic oceanic crust (sediment-starved) and should promote faster slab sinking, slab rollback and convergence velocities (Behr et al., 2022).

## 5   Conclusions

Our load-stepping experiments on powdered aggregates of nominally pure glaucophane provide new insights into the rheologi-
290    cal behavior of sodic amphibole in subduction zones. The mechanical data reveal a clear transition in deformation mechanisms,



with stress exponent values of 2.8 at relatively low stresses— consistent with dislocation creep— to values of 14-19 at relatively high stresses, indicative of dislocation glide. Microstructurally, we document kinking and undulose extinction in the deformed samples, along with a notable change in grain boundary morphology from sharp, linear boundaries in the hydrostatic sample to rounded and lobate grain boundaries in the deformed samples. High internal misorientations in both relict and recrystallized grains further confirm the activation of dislocation glide under high-stress conditions. To quantify these deformation mechanisms, we used the mechanical data to develop two distinct flow laws representing dislocation creep and dislocation glide. Extrapolating these flow laws to natural subduction conditions using deformation mechanism maps, we estimate that coarse-grained sodic amphibole-bearing rocks can deform via dislocation creep at temperatures of 450°C and higher. This aligns with observations of naturally deformed high-pressure rocks, suggesting that dislocation creep in mafic rocks plays a significant role in accommodating strain along the subduction interface. Overall, this study highlights the importance of dislocation-related deformation mechanisms in mafic subduction interface materials and provides new constraints on the strength and viscosity of blueschist-facies rocks. These findings have implications for understanding long-term subduction dynamics, including the potential for strain localization, transient behavior, and the mechanical behavior of subduction interfaces at varying depths and temperatures.

*Author contributions.*



Conceptualization: WMB, LT, LJH Data Curation: LJH, LT, LFGM Formal Analysis: LJH, LT, WMB Software: CM Supervision: LT, WMB Visualization: LJH, LT, WMB Writing- original draft: LJH, LT, WMB Writing- review and editing: LJH, LT, WMB, LFGM, CM

*Competing interests.*

The authors declare that they have no competing interests.

*Acknowledgements.* We thank Remi Lüchinger for making the thin sections, Thomas Mörgeli and Simon Schmid for technical support. This research was funded by an ERC starting grant to Whitney M. Behr (S-SIM, Grant Number 947659).



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
