# Peer review of "Dislocation creep and glide in experimentally deformed glaucophane aggregates"

_EGUsphere, 2025_

## Referee Comment (RC1)

Orléans, the 07th March 2025

Dear Editor, Dear Authors,

I have read in detail the manuscript entitled "Dislocation creep and glide in experimentally deformed glaucophane aggregates" by Lonnie J. Hufford and others. This works aims at constraining, on the basis of 5 deformation experiments, both the microscopic deformation processes and the corresponding macroscopic flow laws. This works is integrated in a larger series of works dedicated to the rheology of the various rocks deforming along the subduction interface.

While rheological contrasts are clearly visible in deep metamorphic rocks from subduction zones, consistent rheological flaw laws are missing, and this work is very relevant in this respect. A difficulty lies in the fact that part of the deformation processes described in this work pertain to the transition between fully plastic deformation and brittle/frictional deformation. For such conditions, the different possible processes operating are multiple (involving fracturing, frictional slip, chemical mass transfer along grain boundaries, plasticity etc…) and very careful observations are necessary.

The present work falls short of such necessary observations, as the documentation of the microstructures is really poor (see full comment below). For example, none of the microstructures invoked (subgrain boundaries, lobate grain boundaries) are well documented, while the SEM pictures are reminiscent of brittle deformation, which is never considered here.

The experimental dataset is also very short: 5 deformation experiments, without any replicate, to characterize 2 different types of deformation mechanisms and cover ranges of variations in temperature and in strain rate. It includes one deformation experiment where the grain size is different from the rest, and the authors, instead of repeating the experiment with the proper grain size, chose to keep it but to exclude the mechanical data from the dataset used to derive the flow laws. The microstructural work is somehow questionable for load-stepping experiments covering two deformation regimes, as it is then impossible to tie microstructures to either one of the deformation regime. So further experiments are necessary, including a few replicates of the experiments shown, constant strain-rate experiments in either one of the regimes to analyze the microstructures, a new "700°C" experiment etc…

The lack of balance between the observations and the interpretation is apparent in the structure of the work: 40 lines and 4 figures of results, to fully describe a complex mechanical behavior, vs. 140 lines and 4 figures for the discussion. A companion paper is evoked, what is hard to understand, given the fact that the present work is missing many data and observations.

As a result, my conclusion is that the work is an interesting, promising experimental work, on an interesting topic, but it is still far from being completed. I suggest to add the missing data (additional experiments, additional observations, all necessary) and to resubmit the work. If a companion paper is submitted in parallel, an idea to consider (I cannot be sure of it because I have had no access to it) might be to merge the two manuscripts.

Best Regards

Hugues Raimbourg

*PS: 18th of March – The editor sent me the link to the companion paper. I checked it quickly, and it seems that there are indeed many data and observations that are relevant to the present manuscript. I do not find it relevant to split the work into two distinct manuscripts, because the result is that the present paper lacks the necessary information to stand on its own.*

*Plus, I found that some part of the text was repeated between the companion paper and the present one. The last part of the companion paper (L420-437) and of the present manuscript (L271-287) are identical but for a few words. That is an additional argument for merging the two papers. But even more, I consider it an ethical and professional misconduct to copy-paste whole sections of the Discussion between distinct papers.*

**Main comments**

1) Microstructures

There is much microstructural characterization that is missing:

-compositional mapping to show whether dissolution-precipitation, or reactions, or any chemical process occurred and played a role in deformation

- EBSD data: Distribution of misorientation angle of low angle/subgrain boundaries are necessary, as they are directly connected to the active slip system. Pole figures should be shown as well, even if they might not be very informative, because the low amount of strain prevents a clear CPO to develop. Grain size distribution

-Comprehensive set of SEM observations, showing the grain boundary geometries (and if present, the lobate GB), the cracks, the grain size etc…

-pictures showing contrasts in deformation microstructures as a function of temperature (using EBSD, SEM, optical microscopy…)

-TEM work on internal defects

The figures that are provided are too few and of too little quality: In Fig. 2, the optical microscopy images should be zoomed to highlight the microstructures to be observed (and please use the same scale bar on all comparable pictures, for example the large views on the left). In Fig.3, the EBSD maps are obscured by horizontal, superimposed "bands". There is no clear demonstration of the widespread occurrence of subgrains, neither lobate grain boundaries in the EBSD maps. In Fig. 4, the contrast in BSE should be increased to be able to show the presence or absence of newly precipitated amphibole.

Furthermore, a problematic shortcoming of the present work is that the microstructures that are shown reflect a combination of different stages, with potentially different deformation mechanisms. In other words, one cannot ascribe the microstructures to the low- or high-stress regime. A more proper approach would be to combine the load-stepping experiments with constant strain-rate experiments, either in low- or high-stress regime, to observe and describe the relevant microstructures and their difference.

2) Mechanical data

Strain-rate stepping experiments using powders in shear geometry: the final strain achieved is 0.8. The initial stage of compaction, simply to transform a powder with a porosity, onto a non-porous solid sample, takes some strain. For example, the very large sets of experiments on granitoids powders in Griggs shear experiments made by Matej Pec (Pec et al., 2012; Pec et al., 2016) show that strain of the order of $\gamma \sim 1.5$ is necessary to compact the powder, close the porosity and reach steady-state microstructures and mechanical behavior. So it is arguable that the present experiments relied on

steady-state microstructures, hence the bulk properties they derived are applicable to creep properties (which is implicitly a steady-state process). Furthermore, it is conceivable that the mechanical measurements performed in the present work are strain-dependent, hence cannot be converted into a flow law. This is why I suggest repeat one or several of these load-stepping experiments, and to apply, after the last, high load step, a step with a low value of load, comparable to the first step, to check that the mechanical values are not affected by strain and the associated microstructural evolution (see the approach in (Ghosh et al., 2024)). Discarding, through experimental evidence, such a strain- or history dependence is essential to support the derivation of a flow law (where by definition strain and history are absent, as steady-state behavior is assumed).

Brittle/semi brittle behavior: The low-temperature and high-strain rates applied in these experiments may lead to frictional deformation in the brittle regime (see the very large stress exponent and mechanical data in (Marti et al., 2017; Pec et al., 2016)). The fine-grained domains of amphibole in Fig. 4 might be tentatively interpreted as grain crushing, consistent with the interpretation as brittle deformation. I am not claiming that it is the case, but great care should be taken in the present work to discriminate plastic processes (such as the formation of subgrains and their evolution into independent, recrystallized grains) from brittle ones. As stated above, a much more careful microstructural work is necessary to fully support the conclusions of the manuscript.

The fit to a flow law is very disappointing for the "glide" part of the dataset. The 700°C sample shows discrepant results, i.e., a much lower strength compared to what is predicted. This is attributed to a larger initial grain size. Fine, that sort of things can happen during sample preparations. But then, why not repeat the experiment with the same powder and the same grain size as the other 3 samples, to have a robust and comparable "700°C" experiments? Instead of that, the authors discard the experiments and fit their glide law with 3, instead of 4 temperature data, and derive a law out of a very narrow range of temperature (hence large uncertainties when extrapolated to different temperatures).

**Line-to-Line comments:**

L84-85: "A modified version of the open source code RIG (https://mpec.scripts.mit.edu/peclab/software/) for the ETH Zürich Griggs apparatus was used to process the mechanical data." : There are several possible corrections to the raw data, in particular concerning the stress values (Heilbronner et al., 2015). Please provide access to a comprehensive description of the routines applied to the raw data.

L132-138: please provide the compositional maps that shows that precipitated material is limited.

L141-143: The presence of subgrains and lobate grain boundaries is really poorly documented.

L150: the interpretation of high stress exponent as dislocation glide is questionable. It might well be the result of frictional deformation in the brittle regime.

Figure 3: The quality of the map is really poor (especially the deformed sample b and d). A surprising feature is that there is hardly any clear plastic feature in the deformed sample, in spite of the large strain (up to 0.8): the inherited grains are not elongated, preferentially orientated parallel to the shear plane etc… This suggests that most of the strain is actually taken up by the compaction of the powder, which raises many questions as to the mechanical relevance of the initial, low-stress load steps.

Fig. 3: The most visible difference between hydrostatic and deformed sample is the abundance, in the hydrostatic sample, of "white", unindexed areas. Do these correspond to porosity, or to really

uninindexed amphibole grains? Finally, one would expect a low degree of internal misorientation in recrystallized domains (Cross et al., 2017). In contrast, a surprising feature of the deformed sample shown in Fig. 3d is that small grains domains, presumably formed by dynamic recrystallization, show actually larger misorientation than inherited grains.

On this figure, please use the same scale bar for hydrostatic and deformed sample.

**References**

Cross, A.J., Prior, D.J., Stipp, M. and Kidder, S. (2017) The recrystallized grain size piezometer for quartz: An EBSD-based calibration. Geophys. Res. Lett. 44, 6667-6674.

Ghosh, S., Stünitz, H., Raimbourg, H., Précigout, J., Di Carlo, I., Heilbronner, R. and Piani, L. (2024) Importance of grain boundary processes for plasticity in the quartz-dominated crust: Implications for flow law. Earth and Planetary Science Letters 640.

Heilbronner, R., Stünitz, H. and Richter, B. (2015) Calibrating the Grigg's' Apparatus using Experiments performed at the Quartz-Coesite Transition American Geophysical Union, Fall Meeting 2015.

Marti, S., Stünitz, H., Heilbronner, R., Plümper, O. and Drury, M.R. (2017) Experimental investigation of the brittle-viscous transition in mafic rocks – Interplay between fracturing, reaction, and viscous deformation. J. Struct. Geol. 105, 62-79.

Pec, M., Stünitz, H. and Heilbronner, R. (2012) Semi-brittle deformation of granitoid gouges in shear experiments at elevated pressures and temperatures. J. Struct. Geol. 38, 200-221.

Pec, M., Stünitz, H., Heilbrönner, R. and Drury, M.R. (2016) Semi-brittle flow of granitoid fault rocks in experiments at mid-crustal conditions. J. Geophys. Res 121, 1677-1705.

---

## Author Comment (AC1)

REPLY TO Reviewer comments #1
Main comments

1) Microstructures
There is much microstructural characterization that is missing:
-compositional mapping to show whether dissolution-precipitation, or reactions, or any chemical process occurred and played a role in deformation
- EBSD data: Distribution of misorientation angle of low angle/subgrain boundaries are necessary, as they are directly connected to the active slip system. Pole figures should be shown as well, even if they might not be very informative, because the low amount of strain prevents a clear CPO to develop. Grain size distribution
-Comprehensive set of SEM observations, showing the grain boundary geometries (and if present, the lobate GB), the cracks, the grain size etc…
-pictures showing contrasts in deformation microstructures as a function of temperature (using EBSD, SEM, optical microscopy…)
-TEM work on internal defects

We completely agree with the reviewer that this type of microstructural analysis is essential to interpreting experimental data. However, unfortunately, the reviewer didn't have timely access to our companion manuscript, which focuses primarily on this aspect-– several experiments presented in the companion manuscript were specifically designed to examine deformation mechanisms and microstructures in the same starting material as a function of increasing strain. That said, we will take the reviewer's suggestion and include more of this kind of data in a revised version of this manuscript, but we note (and as both reviewers also mentioned) that microstructural analysis on load-stepping experiments that record two distinct stages of mechanical behavior can be difficult to interpret. We will also run an additional constant load experiment that deforms only in the dislocation creep field.

The figures that are provided are too few and of too little quality: In Fig. 2, the optical microscopy images should be zoomed to highlight the microstructures to be observed (and please use the same scale bar on all comparable pictures, for example the large views on the left).
Fair point. We will provide more zoomed in microstructures and standardize the scale bars in a revised version.

In Fig.3, the EBSD maps are obscured by horizontal, superimposed "bands". There is no clear demonstration of the widespread occurrence of subgrains, neither lobate grain boundaries in the EBSD maps.
The banding you are referring to is likely related to charging. This EBSD map took over 85 hours to complete and this long duration likely resulted in the banding; however, it did not affect the indexing or data.

In Fig. 4, the contrast in BSE should be increased to be able to show the presence or absence of newly precipitated amphibole.
The contrast on the BSE images is quite high. The BSE images are largely the same shade because there is little to no newly precipitated amphibole (or other phases). In figure 4b, we point out a small area where you can see a small amount of new amphibole and albite. This reaction only appears in the 700°C sample, where it is minimal (<2%). Our companion manuscript discusses these reactions in more detail because in those experiments they proceed to a much greater extent in fine-grained shear zones that were produced at much higher strains.

Furthermore, a problematic shortcoming of the present work is that the microstructures that are shown reflect a combination of different stages, with potentially different deformation mechanisms. In other words, one cannot ascribe the microstructures to the low- or high-stress regime. A more proper approach would be to combine the load-stepping experiments with constant strain-rate experiments, either in low- or high-stress regime, to observe and describe the relevant microstructures and their difference.

We completely agree with the reviewer here, and that is exactly what we did in our companion manuscript, where we presented several constant strain-rate experiments quenched at incrementally increasing strain, with associated detailed microstructural analyses.

2) Mechanical data Strain-rate stepping experiments using powders in shear geometry: the final strain achieved is 0.8. The initial stage of compaction, simply to transform a powder with a porosity, onto a non-porous solid sample, takes some strain. For example, the very large sets of experiments on granitoids powders in Griggs shear experiments made by Matej Pec (Pec et al., 2012; Pec et al., 2016) show that strain of the order of $\gamma \sim 1.5$ is necessary to compact the powder, close the porosity and reach steady-state microstructures and mechanical behavior. So it is arguable that the present experiments relied on steady-state microstructures, hence the bulk properties they derived are applicable to creep properties (which is implicitly a steady-state process).

Detailed SEM imaging of the hydrostatic experiment demonstrates that there was no porosity in the samples at the start of the experiments.  This means the porosity is sufficiently removed during cold- and hot-pressing prior to deformation. This is consistent with many other experiments on various geologic materials that use powdered starting materials in the shear geometry, and do not show porosity at hydrostatic conditions (e.g.  Nachlas and Hirth 2015; Tokle et al. 2023 JSG; Zhao et al 2019; Speciale et al 2020).

The starting material and deformation conditions of course do affect this. Pec et al. 2012 deformed granitic powder at 300 and 500°C, while Pec et al. 2016 deformed granitic powder at 300-600°C. Both of these studies were targeting deformation conditions that promote semi-brittle flow. Given the initial starting powders, temperature, pressure, and strain rate conditions, neither quartz nor feldspar in those samples are expected to activate diffusional processes (for example, normal grain growth)--- therefore it makes sense that porosity in those experiments remained during pressurization and until deformation commenced and comminution occurred within the sample, removing the porosity. By contrast, Tokle et al. (2023) JGR-SE experiments showed diffusion creep in blueschist occurred at similar deformation temperatures, supporting that diffusional processes may be activated during hot-pressing in the load-stepping experiments in this manuscript.

Furthermore, it is conceivable that the mechanical measurements performed in the present work are strain-dependent, hence cannot be converted into a flow law. This is why I suggest repeat one or several of these load-stepping experiments, and to apply, after the last, high load step, a step with a low value of load, comparable to the first step, to check that the mechanical values are not affected by strain and the associated microstructural evolution (see the approach in (Ghosh et al., 2024)). Discarding, through experimental evidence, such a strain- or history dependence is essential to support the derivation of a flow law (where by definition strain and history are absent, as steady-state behavior is assumed).

Dislocation glide or low temperature plasticity experiments often show work hardening (i.e., a strain dependence) and they are developed into flow laws (e.g. Hansen et al. 2019; Mei et al. 2010; Kronenberg et al. 1990; Burdette and Hirth, 2022). We will look closely at our mechanical data to see if we can find any signature of a strain dependence at both low and high stress data and incorporate new findings in a revised manuscript.

The reviewer suggests repeating the load stepping experiments. However, achieving sufficient strain in our shear experiments at the slow strain rates required for dislocation creep in glaucophane took a considerable amount of time. Because dislocation creep in glaucophane had not previously been characterized, we aimed to generate enough data within the dislocation creep regime to establish a robust stress exponent relationship, rather than just identifying the transition between deformation mechanisms. Consequently, the experiments that captured the low-stress regime (n ~ 3) lasted over two weeks at 700°C, and the 600°C experiment ran for 12 weeks. Most published dislocation glide flow laws for geologic materials do not include repeated load or strain rate-stepping experiments (e.g., Mei et al. 2010; Renner et al. 2002; Raterron et al. 2004; Kronenberg et al. 1990; Mares & Kronenberg 1993; Burdette and Hirth 2022). While we acknowledge that repeated loading or stepping can be useful under certain conditions, we do not consider it necessary to address the scientific questions posed by our study.

Brittle/semi brittle behavior: The low-temperature and high-strain rates applied in these experiments may lead to frictional deformation in the brittle regime (see the very large stress exponent and mechanical data in (Marti et al., 2017; Pec et al., 2016)). The fine-grained domains of amphibole in Fig. 4 might be tentatively interpreted as grain crushing, consistent with the interpretation as brittle deformation. I am not claiming that it is the case, but great care should be taken in the present work to discriminate plastic processes (such as the formation of subgrains and their evolution into independent, recrystallized grains) from brittle ones. As stated above, a much more careful microstructural work is necessary to fully support the conclusions of the manuscript.

The fine-grained material referenced in Figure 4 is indeed developed during cold pressing and initial pressurization due to grain crushing— this is a very common consequence of using a powdered starting material. We document this in both this manuscript and the companion manuscript. We also do not interpret it as related to dynamic recrystallization in the dislocation creep regime because strains are not sufficiently high during the n~3 load-stepping steps. We will make these points more clear in the revised manuscript.

As for the high stress exponents and microstructures observed in Marti et al. 2017 and Pec et al. 2016. They paired their high stress exponent relationships with microstructures that supported semi-brittle to brittle deformation behavior. They show the development of highly localized shear bands and strain localization, which we do not observe, as well as amorphous grains, which we also do not observe.

The fit to a flow law is very disappointing for the "glide" part of the dataset. The 700°C sample shows discrepant results, i.e., a much lower strength compared to what is predicted. This is attributed to a larger initial grain size. Fine, that sort of things can happen during sample preparations. But then, why not repeat the experiment with the same powder and the same grain size as the other 3 samples, to have a robust and comparable "700°C" experiments? Instead of that, the authors discard the experiments and fit their glide law with 3, instead of 4 temperature data, and derive a law out of a very narrow range of temperature (hence large uncertainties when extrapolated to different temperatures).

Upon reviewing the microstructures in detail, we discovered that the grain size for the 700°C was notably larger than the other three experiments. Hansen et al. 2019 showed that olivine deforming by LTP is sensitive to grain size where samples with larger grain sizes are weaker than samples that have finer grain sizes, which is also what we observed. We currently do not have enough data to model this grain size effect, therefore we opted to fit the glide flow law to the experimental data of similar grain size, Figure 6b. Based on the reviewer's suggestion, however, we will conduct an additional 700 C experiment at a finer grain size consistent the lower temperature samples.

Line-to-Line comments:
L84-85: "A modified version of the open source code RIG
(https://mpec.scripts.mit.edu/peclab/software/) for the ETH Zürich Griggs apparatus was used to
process the mechanical data." : There are several possible corrections to the raw data, in particular
concerning the stress values (Heilbronner et al., 2015). Please provide access to a comprehensive
description of the routines applied to the raw data.
We have already provided in the text the routine we use to correct the raw data. Additionally, the code
was provided in the supplementary document.

P.S. The Heilbronner et al. (2015) citation the reviewer references is an unpublished AGU abstract and
the presentation itself is not publicly available, nor peer-reviewed. Based on the abstract text, this poster
is about a salt correction that influences the stresses at high strains relative to the area correction.
However, no study that we are aware of has applied this correction. Additionally, this correction is
primarily aimed at general shear experiments conducted to high shear strains (gamma > 3) where the
area correction is thought to overcorrect the stress– our experiments are conducted to low shear strains,
so this correction is not relevant to our study.

L132-138: please provide the compositional maps that shows that precipitated material is limited.
Our companion manuscript that you did not have access to contained several images and EDS maps of
the hydrostatic starting materials (showing very limited reaction textures).  We will also add maps like
these into this manuscript during revision.

L141-143: The presence of subgrains and lobate grain boundaries is really poorly documented.
The documentation of lobate grain boundaries is shown primarily by EBSD (maps in this manuscript, but
also more clearly in maps of our companion manuscript). These samples are >97% glaucophane and
therefore it is quite difficult to visualize grain boundary structure– we also had significant challenges in
indexing due to the fine grain size. We are working on etching the samples to allow for better
documentation of the microstructures in the SEM-BSE. These images will be included in the revised
manuscript.

L150: the interpretation of high stress exponent as dislocation glide is questionable. It might well be the
result of frictional deformation in the brittle regime.
We interpret that the mechanical data represents dislocation glide/LTP, based on a) homogeneous strain
(no localization) within the sample b) high misorientations in the fine grained regions represent high
dislocation densities and c) that the fine grained regions are nominally the same between the hydrostatic
and deformed samples (suggesting no significant comminution).

Figure 3: The quality of the map is really poor (especially the deformed sample b and d). A surprising
feature is that there is hardly any clear plastic feature in the deformed sample, in spite of the large strain
(up to 0.8): the inherited grains are not elongated, preferentially orientated parallel to the shear plane
etc… This suggests that most of the strain is actually taken up by the compaction of the powder, which
raises many questions as to the mechanical relevance of the initial, low-stress load steps.
There is no meaningful amount of porosity (i.e., <<1%) in our hydrostatic sample, therefore strain is
definitely not accommodated by the compaction of the starting powder.

In regards to EBSD, indexing amphibole grains is very difficult and even more difficult when they are
loaded with dislocations, as is the case in the deformed sample. It took us a long time to find a polishing
and run procedure to allow for meaningful indexing of amphibole, especially at small grain sizes.

Fig. 3: The most visible difference between hydrostatic and deformed sample is the abundance, in the hydrostatic sample, of "white", unindexed areas. Do these correspond to porosity, or to really unindexed amphibole grains? Finally, one would expect a low degree of internal misorientation in recrystallized domains (Cross et al., 2017). In contrast, a surprising feature of the deformed sample shown in Fig. 3d is that small grains domains, presumably formed by dynamic recrystallization, show actually larger misorientation than inherited grains.

We are rerunning the EBSD map for the hydrostatic experiment for the revised manuscript.

The finite strains in the dislocation creep regime in these experiments are quite low (~0.04-0.05 gamma), so we do not interpret them as sufficient to produce a lot of dynamic recrystallization (though see our companion manuscript for higher strain experiments conducted at constant strain rate, where we do see DRX). The high misorientations in the grains that the reviewer notes likely reflect an increase in dislocation density produced when the samples enter the dislocation glide regime, which is what we would expect for a sample that deformed by dislocation glide. So the microstructures match/support our interpretation of deformation by glide. We will make these interpretations more clear in the revised manuscript.

On this figure, please use the same scale bar for hydrostatic and deformed sample.

This has been fixed. Thanks.

The last part of the companion paper (L420-437) and of the present manuscript (L271-287) are identical but for a few words.

Thanks to the reviewer for noticing this overlapping text– this was an oversight on our part when moving content from one manuscript to another during revisions to the first manuscript and preparation of this one. We will certainly fix this.

Reviewer References
Cross, A.J., Prior, D.J., Stipp, M. and Kidder, S. (2017) The recrystallized grain size piezometer for quartz: An EBSD-based calibration. Geophys. Res. Lett. 44, 6667-6674.
Ghosh, S., Stünitz, H., Raimbourg, H., Précigout, J., Di Carlo, I., Heilbronner, R. and Piani, L. (2024) Importance of grain boundary processes for plasticity in the quartz-dominated crust: Implications for flow law. Earth and Planetary Science Letters 640.
Heilbronner, R., Stünitz, H. and Richter, B. (2015) Calibrating the Grigg's' Apparatus using Experiments performed at the Quartz-Coesite Transition American Geophysical Union, Fall Meeting 2015.
Marti, S., Stünitz, H., Heilbronner, R., Plümper, O. and Drury, M.R. (2017) Experimental investigation of the brittle-viscous transition in mafic rocks – Interplay between fracturing, reaction, and viscous deformation. J. Struct. Geol. 105, 62-79.
Pec, M., Stünitz, H. and Heilbronner, R. (2012) Semi-brittle deformation of granitoid gouges in shear experiments at elevated pressures and temperatures. J. Struct. Geol. 38, 200-221.
Pec, M., Stünitz, H., Heilbrönner, R. and Drury, M.R. (2016) Semi-brittle flow of granitoid fa

Reply References
-Burdette, E., and Hirth, G., (2022). Creep Rheology of Antigorite…
-Hansen, L., Kumamoto, K.M., Thom, C.A., Wallis, D., Durham, W.B., Goldsby, D.L., Breithaupt, Meyers, C.D., and Kohlstedt, D.L., (2019) Low-Temperature plasticity in olivine…
-Kronenberg, A.K., Kirby, S.H., Pinkston, J., (1990) Basal Slip and Mechanical Anisotropy of Biotite.
-Mares, V.M., and Kronenberg, A.K. (1993) Experimental deformation of muscovite.
-Mei, S., Suzuki, A.M., Kohlstedt, D.L., Dixon, N.A., and Durham, W.B. (2010) Experimental constraints…
-Nachlas, W.O., and Hirth, G. (2015) Experimental constraints on the role of dynamic…
-Raterron, P., Wu, Y., Weidner, D.J., and Chen, J. (2004). Low-temperature olivine rheology …
-Renner, J., Evans, B., and Siddiqi, G. (2002). Dislocation creep of calcite.
-Speciale, P., Behr, W.M., Hirth, G., and Tokle, L. (2020) Rates of Olivine grain growth
-Tokle, L., Hirth, G., and Stünitz (2023) The effect of muscovite…
-Tokle, L., Hufford, L.J., Behr, W.M., Morales, L.F.G., and Madonna (2023) Diffusion Creep of Sodic…
-Zhao, N., Hirth, G., Cooper, R.F., Kruckenberg, S.C., and Cukjati, J. (2019). Low viscosity of mantle…